# Failure of Lactate Clearance Predicts the Outcome of Critically Ill Septic Patients

**DOI:** 10.3390/diagnostics10121105

**Published:** 2020-12-18

**Authors:** Raphael Romano Bruno, Bernhard Wernly, Stephan Binneboessel, Philipp Baldia, Dragos Andrei Duse, Ralf Erkens, Malte Kelm, Behrooz Mamandipoor, Venet Osmani, Christian Jung

**Affiliations:** 1Medical Faculty, Division of Cardiology, Pulmonology and Vascular Medicine, University Hospital Düsseldorf, Heinrich-Heine-University Düsseldorf, 40225 Düsseldorf, Germany; rrbruno@outlook.de (R.R.B.); stephan.binneboessel@med.uni-duesseldorf.de (S.B.); philipp.baldia@med.uni-duesseldorf.de (P.B.); dragos-andrei.duse@med.uni-duesseldorf.de (D.A.D.); ralf.erkens@med.uni-duesseldorf.de (R.E.); malte.kelm@med.uni-duesseldorf.de (M.K.); 2Department of Cardiology, Clinic of Internal Medicine II, Paracelsus Medical University of Salzburg, 5020 Salzburg, Austria; bernhard@wernly.at; 3Division of Cardiology, Department of Medicine, Karolinska Institutet, Karolinska University Hospital, Solna, 171 64 Stockholm, Sweden; 4Cardiovascular Research Institute Düsseldorf (CARID), 40225 Düsseldorf, Germany; 5Fondazione Bruno Kessler Research Institute, 38123 Trento, Italy; bmamandipoor@fbk.eu (B.M.); vosmani@fbk.eu (V.O.)

**Keywords:** sepsis, intensive care, critically ill, lactate, microcirculation

## Abstract

Purpose: Early lactate clearance is an important parameter for prognosis assessment and therapy control in sepsis. Patients with a lactate clearance >0% might differ from patients with an inferior clearance in terms of intensive care management and outcomes. This study analyzes a large collective with regards to baseline risk distribution and outcomes. Methods: In total, 3299 patients were included in this analysis, consisting of 1528 (46%) ≤0% and 1771 (54%) >0% patients. The primary endpoint was intensive care unit (ICU) mortality. Multilevel logistic regression analyses were used to compare both groups: A baseline model (model 1) with lactate clearance as a fixed effect and ICU as a random effect was installed. For model 2, patient characteristics (model 2) were included. For model 3, intensive care treatment (mechanical ventilation and vasopressors) was added to the model. Models 1 and 2 were used to evaluate the primary and secondary outcomes, respectively. Model 3 was only used to evaluate the primary outcomes. Adjusted odds ratios (aORs) with respective 95% confidence intervals (CI) were calculated. Results: The cohorts had no relevant differences regarding the gender, BMI, age, heart rate, body temperature, and baseline lactate. Neither the primary infection focuses nor the ethnic background differed between both groups. In both groups, the most common infection sites were of pulmonary origin, the urinary tract, and the gastrointestinal tract. Patients with lactate clearance >0% evidenced lower sepsis-related organ failure assessment (SOFA) scores (7 ± 6 versus 9 ± 6; *p* < 0.001) and creatinine (1.53 ± 1.49 versus 1.80 ± 1.67; *p* < 0.001). The ICU mortality differed significantly (14% versus 32%), and remained this way after multivariable adjustment for patient characteristics and intensive care treatment (aOR 0.43 95% CI 0.36–0.53; *p* < 0.001). In the additional sensitivity analysis, the lack of lactate clearance was associated with a worse prognosis in each subgroup. Conclusion: In this large collective of septic patients, the 6 h lactate clearance is an independent method for outcome prediction.

## 1. Introduction

Sepsis is one of the most common and deadly diseases worldwide. It is characterized by a life-threatening “disproportionate” immune response to infection, with high mortality [1,2]. The long-term morbidity of survivors is also a major public health problem [3,4,5,6]. In daily clinical practice, serum lactate elevation is among the most often used parameters for outcome prediction. Lactate metabolism had been introduced as one of the most promising approaches for individualized treatment in septic patients [7,8]. Thanks to point-of-care testing, lactate measurement is now generally available [9]. The pathophysiology is still the subject of lively debate; although it is well known that high lactate indicates a patient at risk, we do not know if elevated lactate represents the consequence of, or the physiological response to, critical illness. Historically, lactate has been interpreted as a marker of anaerobic glycolysis following tissue hypoxia. More recent studies show, however, that lactate is a central key in glucose metabolism, and is not the result of anaerobic glycolysis [10]. From a pathophysiological point of view, hyperlactatemia could result from a deficit in oxygen delivery or impaired oxygen extraction [11], peripheral shunting [12], and increased endo- or exo-genous adrenergic stimulation [13]. Importantly, serum lactate is influenced by many other factors (catecholamine-use in septic shock, alkalosis-induced increased glucose metabolism, lactate buffered continuous hemofiltration, liver dysfunction, and lung lactate production) [14,15]. In acutely ill patients, it is likely hardly possible to recognize the weight of the individual factors. 

Lactate clearance was introduced to reduce the influence of these confounders. This approach aims at the ability of the organism to metabolize lactate faster than it is produced. Lactate clearance after 24 h predicts intensive care unit (ICU) mortality quite well [16]. However, the early stage of sepsis often develops quickly. Therefore, clearance after 6 h might better reflect the “golden hours” of sepsis [17]. Several guidelines use lactate clearance as a target to guide therapy, but these recommendations are based on supporting evidence of a low quality. In fact, most of the studies were single-centered and analyzed relatively few patients. 

In this study, we use the eICU Collaborative Research Database ICU database, including over 200,000 admissions from different hospitals [18]. 

This analysis aims to compare patients with sepsis, with and without lactate clearance, in this large collective in terms of baseline risk distribution, management, and outcomes.

## 2. Methods

### 2.1. Study Subjects

The eICU Collaborative Research Database comprises a multi-center intensive care unit (ICU) database, which includes over 200,000 admissions from 335 ICUs of 208 hospitals throughout the USA for the years 2014 and 2015 [18]. The database is distributed under the Health Insurance Portability and Accountability Act (HIPAA) safe harbor provision. Baseline characteristics and organ support on day one were extracted. To detect septic patients, sepsis was identified based on the method established by Angus et al., who successfully used billing codes [19]. In addition, management strategies, defined as the use of vasopressors and mechanical ventilation, the (predefined) type of primary infection site, and the ethical background, were extracted. We included only patients with an initial lactate concentration >2.0 mmol/L in this analysis so as to prevent confounding lactate dynamics within the normal range.

### 2.2. Statistical Analysis

Continuous data points are expressed as median ± interquartile range. Differences between independent groups were calculated using Mann–Whitney U-test accordingly. Categorical data are expressed as numbers (percentage). A Chi-square test was applied to calculate the univariate differences between groups. Lactate clearance was defined as serum lactate at admission minus lactate after 6 h, divided by lactate at admission multiplied by 100. Thus, a positive value indicates a fall in serum lactate and a negative value signifies rising serum lactate. The primary exposure was serum lactate clearance (≤0% or >0%), and the primary outcome was ICU-mortality. The secondary outcomes were the management strategies, mechanical ventilation, and vasopressor use. We used three sequential random effect multilevel logistic regression models to assess the impact of lactate clearance on ICU mortality. A baseline model with lactate clearance as a fixed effect and ICU as a random effect (model 1) was installed. Second, patient characteristics (creatinine, age, sepsis-related organ failure assessment (SOFA) score, BMI, infection source, ethnics; model 2) were added to model 1. Third, management strategies (model 3) were added to model 2. Models 1 and 2 were used to evaluate the primary and secondary outcomes, respectively. Model 3 was only used to evaluate the primary outcomes. Adjusted odds ratios (aOR) with respective 95% confidence intervals (CI) were calculated. For the sensitivity analysis, we analyzed only patients with creatinine above 2.0 mg/dL (arbitrary cut-off), lactate above 2.0 mmol/L (arbitrary cut-off), age above 65 years (arbitrary cut-off), SOFA >10 (arbitrary cut-off), a heart rate above 110 beats per minute, and the management strategies used (vasopressor use and mechanical ventilation). All of the tests were two-sided, and a *p*-value of <0.05 was considered statistically significant. Stata 16 was used for all of the statistical analyses. 

## 3. Results

In 3299 patients from the database, sufficient data on lactate kinetics were available. These patients could be included. In total, 1528 (46%) without and 1771 (54%) septic patients with a lactate clearance greater 0% were investigated in this study. Table 1 summarizes the baseline characteristics and risk distribution of the unadjusted cohort. The cohorts had no relevant differences regarding gender, BMI, age, heart rate, body temperature, the baseline lactate. Neither the primary infection focuses nor the ethnic background differed between both groups. In both groups, the most common infection sites were of pulmonary origin, the urinary tract, and the gastrointestinal tract. Patients with a lactate clearance greater 0% were less sick in terms of SOFA, had a lower baseline creatinine, and lower lactate after 6 h. The group unable to clear lactate more often received invasive ventilation and catecholamines (Table 2). 

The patients in the no clearance group received similar total amounts of fluid and amounts of fluid per bodyweight within the first 24 h as the patients with lactate clearance. In addition, the rate of patients with a fluid intake >30 mL/kg/h in the first 24 h was similar between the two groups (Figure 1).

After adjustment for the ICU cluster as a random effect (model 1), the difference between both groups regarding the use of mechanical ventilation (aOR 0.51 95% CI 0.44–0.60, *p* < 0.001) and vasopressors (aOR 0.58 95% CI 0.50–0.69; *p* < 0.001) persisted (Table 2). In the next step, patient-specific confounders were added to the model (model 2). After adjustment, patients with a lactate clearance greater than 0% still evidenced a lower odd for both the use of mechanical ventilation (aOR 0.78 95% CI 0.70–0.87; <0.001) and vasopressors (aOR 0.87 95% CI 0.79–0.97; 0.01; Table 2).

The ICU mortality differed between both groups. Patients without a lactate clearance greater 0% demonstrated an ICU mortality of 32%, while patients who cleared lactate evidenced lower mortality (14%; Figure 2). This difference remained significant after the following adjustments: After the adjustment in model 1 for the ICU cluster as a random effect, there was a significant reduction for ICU mortality in patients with a lactate clearance greater 0% (aOR 0.34 (95% CI 0.28–0.40; *p* < 0.001). Model 2 for patient-specific factors (creatinine, SOFA, BMI, age, ethnics, infection focus, and heart rate) resulted in an aOR of 0.42 (95% CI 0.34–0.52; *p* < 0.001). Model 3, which added management strategies, lead to an aOR of 0.43 (95% CI 0.36–0.53; *p* < 0.001).

The sensitivity analysis (Figure 3) showed that the lack of lactate clearance was associated with a worse prognosis in each subgroup.

## 4. Discussion

This retrospective multicenter analysis confirms the high prognostic relevance of 6-h lactate clearance. High lactate levels are predictive of a bad course, this is no secret. However, it is equally well known that the correct interpretation of the values is the subject of ongoing discussions. In particular, “lactate-controlled therapy” has not yet been able to prove a thorough superiority. Historically, it was assumed that serum lactate is primarily the result of cellular hypoperfusion at a microcirculation level. Accordingly, it was assumed that local oxygen deficiency leads to increased glycolysis with a consecutive increase in lactate [20]. A further problem is that a considerable part of our knowledge about lactate came from sports physiology, and not from intensive care medicine [14].

This study now provides a valuable new input—the inability to break down lactate is a predictor of ICU mortality, independent of disease severity. This robust association of lactate clearance and mortality remains even after the correction of possible confounders and in extensive sensitivity analyses.

This finding fits the current clinical discourse: the current concept for understanding is based on the “lactate shuttle”. In short, this paradigm follows the idea that all glucose molecules entering the cytoplasm are metabolized to lactate. Lactate finally gets oxidized to CO_2_ and water. Once this oxidative capacity is overhauled by lactate production, the excess lactate leaves the cell and enters the plasma [21]. This lactate can function as a substrate in highly oxidative cells (heart and brain) or can contribute to gluconeogenesis (liver and kidney). Of note, most organs (primarily the liver) increase their rate of lactate oxidation in the functioning metabolic units with the lactate input. Thus, they “clear” the blood from the circulating lactate. When the rate of lactate production in the nonfunctioning metabolic units meets the rate of lactate oxidation of the metabolically active functioning units, the plasma concentration reaches a plateau. These pathophysiological concepts illustrate the importance of lactate clearance as a key factor in critically ill metabolism and its interpretation.

In the present study, men were slightly over-represented in the group of patients with lactate clearance. The question of the extent to which there are gender-specific differences in this collective has been investigated in detail by Wernly et al. In the synopsis of all of the findings, no difference between men and women in 17,146 critically ill patients in intensive care patients from the eICU database could be found [22].

Liver function has often been suggested as a limiting factor in the assessment of lactate metabolism. Ha et al. analyzed, in this context, 770 septic patients with liver function disorders. They found that lactate clearance after 6 (and 24) h remained associated with hospital mortality, even after adjustment for potential confounding factors. Of note, 24 h clearance evidenced a higher discriminatory power [23]. Similar observations could be found for metformin users. Park et al. investigated 71 patients in a propensity-matched analysis. Despite elevated lactate levels on admission in metformin users, they did not find any significant difference in lactate levels, lactate clearance, and normalization over the initial 24 h [24].

Our findings are consistent with a metanalysis by Zhang et al. [25]. In ICU patients, they found a sensitivity of 0.83 (95% CI, 0.67–0.92) and a specificity of 0.67 (95% CI, 0.59–0.75) for the lactate clearance predicting the mortality rate. This is supported by Innocenti et al., who enrolled 268 patients suffering from sepsis. In their cohort, a 6 h lactate clearance greater than 10% was associated with improved survival [26]. Similarly, Arnold et al. analyzed 166 septic patients with presumed sepsis. Lactate non-clearance only occurred in 9%, but mortality was 60% for lactate non-clearance versus 19% for lactate clearance [27]. The same observation was done for children. Kumar et al. found that a 6 h lactate clearance below 16.4% was associated with high mortality in 140 children [28]. Accordingly, Kramer et al. analyzed 392 critically ill patients in the emergency department. They found that lactate clearance after 6 h was correlated with their survival after 30 days [29]. At this point, it should be noted that in our study, both groups exhibited pathological lactate values (defined as >2 mmol/L) at baseline, and the baseline values did not differ in a significant way. Neither could we find possible dilution effects; the fluid balance of the first 24 h of both groups did not differ.

However, the superiority of lactate kinetics over lactate values is not clear. Recently, Fürnau et al. found that the lactate value after 8 h was superior for the outcome prediction of patients in cardiogenic shock compared with the baseline lactate or lactate kinetics [30].

Interestingly, lactate clearance might also offer additional information beyond the prediction of mortality. In 93 patients, Jung et al. found that after cardiac arrest, the 6 h lactate clearance was the only parameter that was correlated with a good neurologic outcome [31]. A 6-h lactate clearance of ≤13% could be identified as an optimal cut-off. Of note, in this cohort, it failed at predicting mortality. In a similar setting, Mizutani et al. identified 6 h lactate kinetics as the most important predictor of in-hospital mortality in patients treated with extracorporeal cardiopulmonary resuscitation after cardiac arrest [32]. 

There are other parameters in critically ill patients that are associated with tissue hypoxia and organ failure. Metabolic acidosis can occur independently from hyperlactatemia. The combinations of both parameters can further increase the predictiveness [6]. Adding central venous oxygen saturation might help to distinguish between an oxygen transport deficit from an oxygen utilization deficit [10].

To date, lactate clearance protocols in randomized clinical trials have not shown any improvement in survival [33]. Other methods aim to assess oxygen deficit in the tissue. ANDROMEDA-SHOCK, for example, compares a capillary-refill-based regime versus lactate clearance to guide therapy in septic patients [34]. There is still controversy about the interpretation of the data from this study [35]. Another promising approach is the investigation of sublingual microcirculation using Sidestream Dark Field Imaging (SDF) [36]. In the early stages of shock, there is currently no clear benefit for SDF [37]. 

Currently, the measurement of lactate clearance remains the “gold standard” for the assessment of the microcirculation in critically ill patients.

## 5. Limitations

This retrospective data analysis has some major limitations. We have no detailed information about microbiological diagnostics. Therefore, it is not possible to distinguish between nosocomial or community-acquired, multi-resistant or sensitive Gram-positive or Gram-negative pathogens or positive bacteremia or sterile blood cultures. However, these limitations are compensated by various measures. The primary focus of infection is a strong predictor of outcome in septic patients [38]. This information could be extracted and included in our large multivariable regression analysis. Another problem is that in this study, we only know the acute disease state in the form of SOFA scores, but no information about the comorbidities. However, this is contrasted by the high number of patients included, so the disadvantages should be outweighed. Lastly, information on fluid amounts was available only for a proportion of the patients (n = 1549), a limitation that could make this analysis prone to selection bias. We abstained from imputation because of the high proportion and decided instead to state this limitation clearly, not only in the Limitations but also in the Results and Figure 1.

## 6. Conclusions

This large retrospective analysis evidenced that a lactate clearance after 6 h greater 0% is a stable and reliable marker to predict the outcome of critically ill septic patients, independently from SOFA and the need for organ support. Further prospective clinical trials are urgently needed.

## Figures and Tables

**Figure 1 diagnostics-10-01105-f001:**
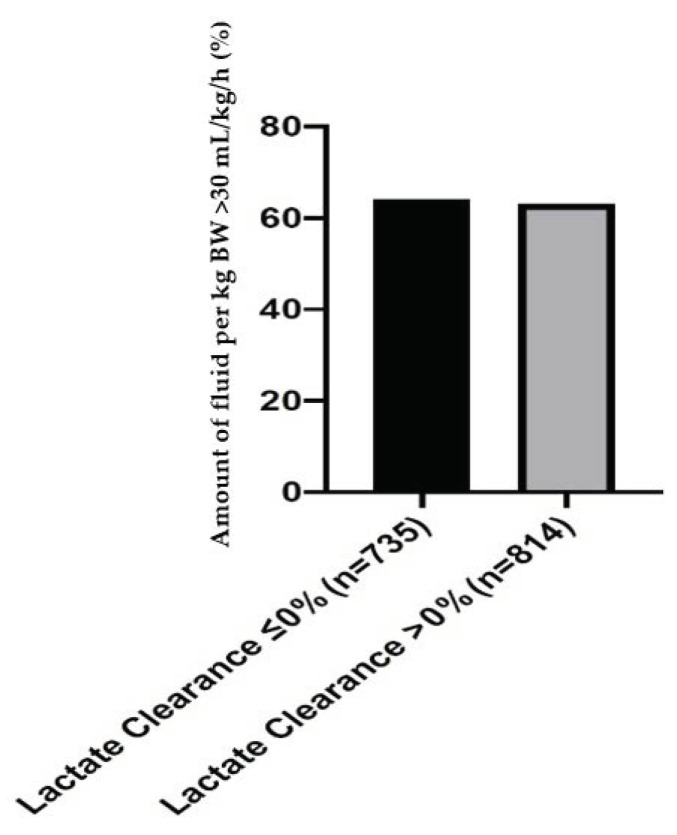
Percentage of patients with a fluid intake greater than 30 mL/kg/h in the first 24 h for patients with a 6-h lactate clearance ≤0%versus >0% in [%].

**Figure 2 diagnostics-10-01105-f002:**
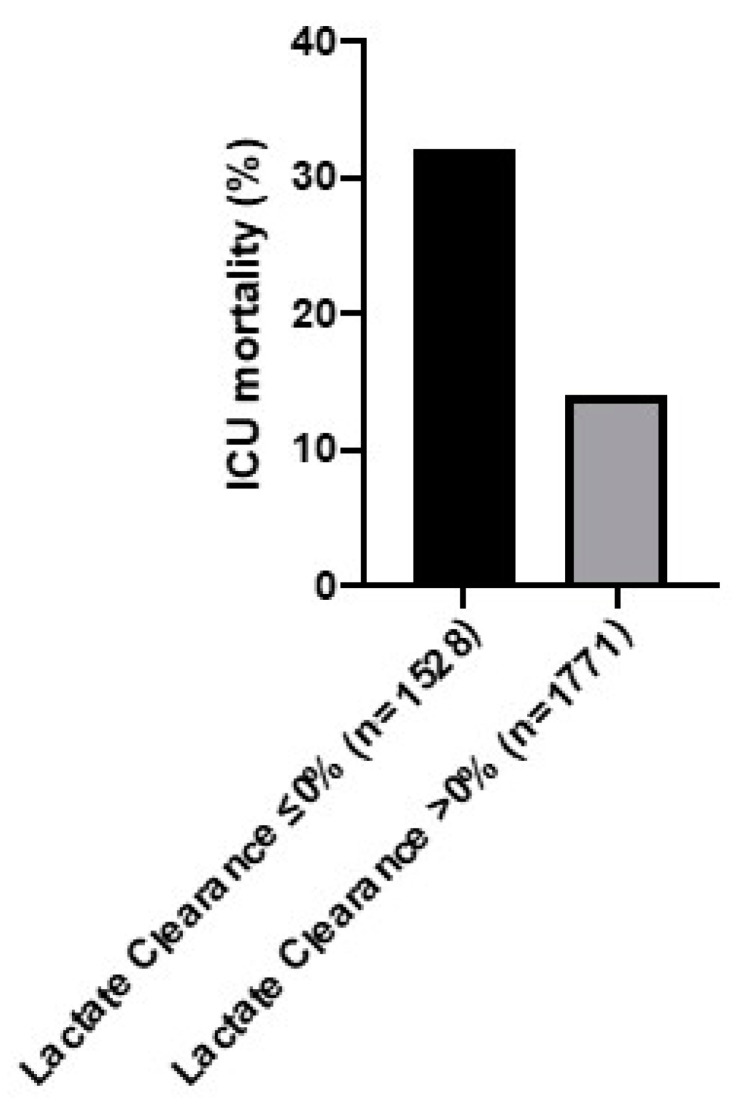
ICU mortality for patients with a 6-h lactate clearance ≤0%versus > 0% in [%].

**Figure 3 diagnostics-10-01105-f003:**
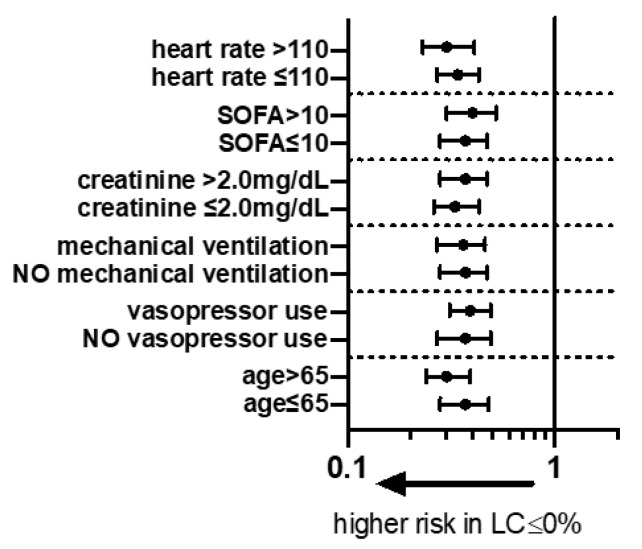
Forest plot of odds ratio (OR) patients with a 6-h lactate clearance ≤0%versus > 0 for different subgroups according to model 1. SOFA—sepsis-related organ failure assessment.

**Table 1 diagnostics-10-01105-t001:** Baseline characteristics in patients with a 6 h lactate clearance ≤0% versus >0%.

	Lactate Clearance ≤ 0%	Lactate Clearance > 0%	
	*n* = 1528	*n* = 1771	*p*-Value
Male sex	753 (49)	947 (54)	0.02
BMI	27 (10)	27 (10)	0.27
Age (years)	67 (21)	66 (21)	0.56
Age > 65 years	802 (53)	923 (52)	0.83
SOFA score	9 (6)	7 (6)	<0.001
SOFA > 10	595 (39)	436 (25)	<0.001
Heart rate >110 bpm	567 (39)	620 (37)	0.20
Body temperature >38 °C	203 (14)	228 (14)	0.76
Creatinine (mg/dL)	1.80 (1.67)	1.53 (1.49)	<0.001
Creatinine > 2.0 mg/dL	655 (44)	632 (36)	0.001
Lactate			
Baseline (mmol/L)	3.40 (2.90)	3.60 (2.60)	0.11
at 6 hours (mmol/L)	4.00 (4.00)	2.20 (1.80)	<0.001
Focus			
UTI	273 (18)	369 (21)	0.03
Pulmonary	555 (36)	607 (34)	0.22
GI	268 (18)	306 (17)	0.84
Cutaneous	90 (6)	122 (7)	0.24
Unknown	215 (14)	248 (14)	0.96
Other	123 (8)	113 (6)	0.06
Gynecologic	4 (<1)	6 (<1)	0.67
*Ethnic*			
Caucasian	1165 (76)	1376 (78)	0.32
AfricanAmerican	172 (11)	180 (10)	0.31
Hispanic	59 (4)	66 (4)	0.84
Asian	28 (2)	34 (2)	0.85
Native American	16 (1)	19 (1)	0.94
Other	88 (6)	96 (5)	0.67
Length of stay (h)	66 (115)	67 (95)	0.48
Fluid management in first 24 h			
Total amount of fluids (mL); median (IQR)	3358 (3860)	3375 (3377)	0.90
Amount of fluid per kg bodyweight; median (IQR)	43 (59)	42 (50)	0.77
Amount of fluid per kg BW >30 mL/kg/h; n (%)	454 (64)	493 (63)	0.73

SOFA—sepsis-related organ failure assessment; BMI—body mass index; UTI—urinary tract infection; GI—gastrointestinal.

**Table 2 diagnostics-10-01105-t002:** Associations of primary exposure (lactate clearance > 0%) with mortality and management strategies in three multilevel logistic regression models.

	Crude Events				
	Lactate Clearance ≤ 0% (n = 1528)	Lactate Clearance > 0% (n = 1771)	Model 1	Model 2	Model 3
	n (%)	n (%)	aOR (95% CI, *p*-value)	aOR (95% CI, *p*-value)	aOR (95% CI, *p*-value)
ICU mortality	488 (32)	242 (14)	0.34 (0.28–0.40; <0.001)	0.42 (0.34–0.52; <0.001)	0.43 (0.36–0.53; <0.001)
Management					
Mechanical ventilation	617 (40)	503 (28)	0.51 (0.44–0.60; <0.001)	0.85 (0.70–1.03; 0.01)	
Vasopressor use	817 (54)	653 (37)	0.58 (0.50–0.69; <0.001)	0.69 (0.57–0.83; <0.001)	

Model 1—Intensive care unit (ICU) cluster as a random effect; Model 2—model 1 plus patient level (creatinine, SOFA, BMI, age, ethnics, infection focus, and heart rate); Model 3—model 2 plus management strategies (mechanical ventilation and vasopressor use); SOFA—sepsis-related organ failure assessment; BMI—body mass index.

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
