# Peer review of "Failure of Lactate Clearance Predicts the Outcome of Critically Ill Septic Patients"

_diagnostics, 2020, doi:10.3390/diagnostics10121105_

Round 1

Reviewer 1 Report

Bruno et al. present a manuscript investigating on the rate of lactate clearance as a clinical predictor for critically ill patients. There is a major limitation of study I can envision in addition what authors already included as limitations. Authors don't mention whether serum lactate levels of subjects were within normal range. For every subject having serum lactate levels at lower end of the normal range any subsequent measurements will result in lactate clearance of 0% or even a negative slope may be evident. Therefore, authors must include serum lactate levels (both baseline and L6). This maybe a confounder of the study. 

Besides this limitation authors need to get their manuscript professionally edited by a native English speaker. 

Author Response

  1. Bruno et al. present a manuscript investigating on the rate of lactate clearance as a clinical predictor for critically ill patients. There is a major limitation of study I can envision in addition what authors already included as limitations. Authors don't mention whether serum lactate levels of subjects were within normal range. For every subject having serum lactate levels at lower end of the normal range any subsequent measurements will result in lactate clearance of 0% or even a negative slope may be evident. Therefore, authors must include serum lactate levels (both baseline and L6). This maybe a confounder of the study. 

Answer: The reviewer points out a very important point: The level of the baseline value is decisive for the calculation of the 6h clearance. In the revised manuscript, we, therefore, clarify that only patients with a baseline defined as pathologically lactate > 2 mmol/L were included in the study. Table 1 shows the corresponding baseline and 6h values. The baseline values between both groups are not statistically different. We also excluded that patients with lactate clearance received volume significantly earlier and more volume (Table 1, Figure 3). The amount of parental volume substitution does not differ either cumulatively or per patient in ml/kg body weight. The number of patients who received more than 30 ml/kg body weight/ hour is also the same in both groups. We hope that these additional explanations underline the value of our results. We could already show with our preliminary work that the 24h-lactate clearance is a strong predictor for the mortality of intensive care patients (Intensive Care Med, 2019 Jan;45(1):55-61. doi: 10.1007/s00134-018-5475-3). But especially in the early phase, the decisive course must be set for septic patients, and this is where the present study comes in: The 6h Clearance also offers the possibility to identify patients at risk early and accurately.

  1. Besides this limitation authors need to get their manuscript professionally edited by a native English speaker.

Answer: We followed your advice. The entire manuscript has been thoroughly and carefully checked and improved.

Reviewer 2 Report

  1. The representation if data is not good, people will not get interest with the way of presentation.
  2. data is not enough to convince the conclusion of the study. The author should include kore expts here

Author Response

  1. The representation if data is not good, people will not get interest with the way of presentation. data is not enough to convince the conclusion of the study. The author should include kore expts here

Answer: We apologize that the manuscript could not arouse your interest. We have revised large parts of the introduction and the discussion linguistically and conceptually.

Round 2

Reviewer 1 Report

Authors addressed the major issue raised by the reviewer. However, there are still issues with the manuscript: 

1. There is a discrepancy in Table 1. In lactate clearance of <0% group there were 1528 subjects as mentioned in the table and in the text. However, authors demonstrate that within the same group there were 1995 male subjects. 

2. Also, there were obviously significantly more male subjects in lactate clearance of <0% group (reported p value of 0.03). Can this be a confounder? 

3. Another potential confounder is the creatinine, the serum levels of which were significantly higher in the group with higher mortality. Since creatinine is a direct measure of renal filtration function, a more thorough discussion has to be made around this finding. 

Author Response

Reviewer 1:

Authors addressed the major issue raised by the reviewer. However, there are still issues with the manuscript: 

  1. There is a discrepancy in Table 1. In lactate clearance of <0% group there were 1528 subjects as mentioned in the table and in the text. However, authors demonstrate that within the same group there were 1995 male subjects. 

Answer: Please excuse this mistake due to a typing error. We have corrected the relevant passage and carefully checked the entire table.

  1. Also, there were obviously significantly more male subjects in lactate clearance of <0% group (reported p value of 0.03). Can this be a confounder? 

Answer: The influence of gender on ICU prognosis is indeed an intriguing question. We have intensively addressed this question in a further analysis of the eICU database but could not find a gender-specific effect. In the revised manuscript, we have now discussed this aspect and provided an appropriate reference:

“In the present study, men were slightly over-represented in the group of patients with lactate clearance. The question of the extent to which there are gender-specific differences in this collective has been investigated in detail by Wernly et al. In the synopsis of all findings, no difference between men and women in 17,146 patients critically ill intensive care patients from the eICU database could be found 22.”

  1. Another potential confounder is the creatinine, the serum levels of which were significantly higher in the group with higher mortality. Since creatinine is a direct measure of renal filtration function, a more thorough discussion has to be made around this finding.

Answer: This is a very good observation, so we added creatinine as a confounder on uptake to the logistic regression model (model-2). The result has not changed significantly. In the revised manuscript you will now find the modified table.

Reviewer 2 Report

The author didn’t work of representation as suggested in the first review. Doesn’t fulfill the interest of a reader at all.

Author Response

Reviewer2:

The author didn’t work of representation as suggested in the first review. Doesn’t fulfill the interest of a reader at all.

Answer: Reviewer #2 mentioned that we failed to address the comments raised in the first round of reviews.

For a better overview we copy the complete first review of our full paper here:

“The representation if data is not good, people will not get interest with the way of presentation. data is not enough to convince the conclusion of the study. The author should include kore expts here”

This can be divided into two parts:

1) The reviewer suggests changing the way of data presentation without specific comments or suggestions. The way of presenting the data is a common standard, the way to calculate lactate clearance and analyzing it is state of the art. Statistical analysis is in our eyes appropriate. Reviewer #1 confirms this, although the suggested improvements have been incorporated.

2) The second part refers to the suggestion that we should involve core experts. We do not understand this as a constructive comment. However, we would like to point out that the current state-of-the-art paper in clinical practice (Masyuk et al., 2019, Intensive Care Medicine,  IF:18) is written by us.

We acknowledge that we obviously do not convince reviewer #2, however, we do not feel that we missed addressing specific comments.

Round 3

Reviewer 1 Report

Authors addressed all the comments this reviewer had. 

Reviewer 2 Report

no further comments